# TAGMOL: Target-Aware Gradient-guided Molecule Generation

**Vineeth Dorna** [* 1 2]  **D. Subhalingam** [* 1]  **Keshav Kolluru** [1]  **Shreshth Tuli** [1 3]  **Mrityunjay Singh** [1]  **Saurabh Singal** [1]
**N. M. Anoop Krishnan** [4]  **Sayan Ranu** [5]

## Abstract

3D generative models have shown significant promise in *structure-based drug design (SBDD)*, particularly in discovering ligands tailored to specific target binding sites. Existing algorithms often focus primarily on ligand-target binding, characterized by binding affinity. Moreover, models trained solely on target-ligand distribution may fall short in addressing the broader objectives of drug discovery, such as the development of novel ligands with desired properties like druglikeness, and synthesizability, underscoring the multifaceted nature of the drug design process. To overcome these challenges, we decouple the problem into molecular generation and property prediction. The latter synergistically *guides* the diffusion sampling process, facilitating guided diffusion and resulting in the creation of meaningful molecules with the desired properties. We call this guided molecular generation process as TAGMOL. Through experiments on benchmark datasets, TAGMOL demonstrates superior performance compared to state-of-the-art baselines, achieving a 22% improvement in average Vina Score and yielding favorable outcomes in essential auxiliary properties. This establishes TAGMOL as a comprehensive framework for drug generation. The code is available at https://github.com/moleculeai/TAGMol.

## 1. Introduction

The presence of molecular data featuring 3D spatial information has paved the way for the complex field of *structure-based drug design (SBDD)* (Anderson, 2003). The advent of generative AI for molecules has accelerated the rational drug-design—in contrast to the traditional Edisonian trial-and-error approach—with the goal of creating drug-like molecules in 3D space that effectively bind to specific targets. Specifically, deep generative models, such as those proposed by Luo et al. (2021); Liu et al. (2022); Peng et al. (2022), autoregressively generate atoms and bonds, while Zhang et al. (2023) generate motifs. Despite their progress, the performance of autoregressive models heavily relies on the order of generation, as they condition on previously generated atoms, which can lead to error propagation. Alternatively, diffusion models (Sohl-Dickstein et al., 2015; Ho et al., 2020) overcome this limitation by conditioning upon all the atoms simultaneously and there by efficiently generating realistic molecules that demonstrate stronger binding affinities with their intended targets.

The effectiveness of these generative models is heavily reliant on how well training datasets — consisting of protein-ligand complexes — align with desired properties such as binding affinity, drug-like topological features, synthetic accessibility (ease of synthesis), to name a few. For instance, CrossDocked 2020 (Francoeur et al., 2020), a widely-used training dataset for SBDD tasks, predominantly includes complexes with moderate binding affinities. Consequently, models trained solely on such datasets may yield sub-optimal molecules, intrinsically tying their success closely to the dataset's quality. Moreover, drug generation is a multi-faceted process that encompasses not only binding affinity but also a range of other desired properties. Compiling a comprehensive dataset that encompasses a broad range of desired properties poses significant challenges, particularly associated with the high computational costs of assessing a suite of properties. Additionally, refining datasets to meet specific quality constraints can drastically reduce the volume of usable training data. As the number of constraints grows, the likelihood of finding samples that satisfy all these criteria becomes increasingly difficult, further exacerbating the situation. Consequently, generative models trained on lower-quality, but high-volume, data may uninten-

---

[*]Equal contribution  [1]Molecule AI, New Delhi, India [2]Manning College of Information and Computer Sciences, University of Massachusetts Amherst, Massachusetts, USA [3]Happening Technology Ltd, London, England [4]Department of Civil Engineering, Indian Institute of Technology Delhi, New Delhi, India [5]Department of Computer Science and Engineering, Indian Institute of Technology Delhi, New Delhi, India. Correspondence to: Vineeth Dorna <vineeth.dorna@moleculeai.com>, D. Subhalingam <subhalingam.d@moleculeai.com>.

*Accepted at the 1st Machine Learning for Life and Material Sciences Workshop at ICML 2024*. Copyright 2024 by the author(s).

tionally capture suboptimal signals, leading to diminished performance in the context of drug discovery. This situation prompts an exploration into how, during the denoising phase of a generative model, we can effectively introduce desired signals while ensuring meaningful reconstruction.

To address these challenges, we introduce TAGMOL– (Target-Aware Gradient-guided Molecule Generation), wherein we decouple molecular generation and property prediction. We start by training a time-dependent *guide* model that predicts properties from inputs with noise levels similar to those in the base diffusion model. Crucially, we turn the challenge of using inferior quality data, i.e., the property of interest is well spread with the inclusion of suboptimal values in the dataset, to our strength for robust *guide* training. Inspired by classifier guidance in diffusion models (Dhariwal & Nichol, 2021), we use the gradient of *guide* to direct the latent space during the diffusion sampling process, ensuring the reconstructed molecules possess the targeted properties. In the sampling phase, we harness the strengths of both the generative model and the *guide*. This interactive dynamic enables us to explore regions with superior properties while simultaneously denoising the latent space to generate diverse molecules. While gradient guidance is a well-explored concept in the drug discovery domain, many existing methods overlook target awareness (Bao et al., 2022) or fail to integrate 3D structure (Eckmann et al., 2022; Stanton et al., 2022; Lee et al., 2023). In contrast, our approach seeks to simultaneously optimize for both target-aware and molecular properties in 3D space. Furthermore, to address real-world scenarios where multiple property constraints exist, we train separate *guides* for each property, subsequently employing them to steer the diffusion process effectively.

Overall, the key contributions of our work are as follows.

- **Reformulation of the drug-discovery problem:** We reformulate the problem of drug generative modeling moving beyond the myopic lens of optimizing binding activity. The need to optimize other properties of interest, even when these signals are not adequately present in the train set, is explicitly coded into our problem formulation.

- **Algorithm design:** We design a novel generative process, called TAGMOL, which jointly leverages the signals from two different components: an *SE(3) equivariant graph diffusion model* to mimic geometries of the train set, and a *multi-objective guide model*, empowered by an SE(3) invariant GNN, to steer the exploration region of diffusion sampling towards the property of interest by leveraging gradients.

- **Rigorous empirical evaluation:** We demonstrate that our model achieves 22% improvement in average Vina Score, all the while being guided by considerations of binding affinity and crucial pharmacological properties such as QED and SA.

## 2. Problem Statement

Generative models for SBDD aim to create ligand molecules that effectively bind to specific protein binding sites. A protein binding site is characterized by a set of atoms, denoted by $\mathcal{P} = \{(\mathbf{x}_i^{\mathcal{P}}, \mathbf{v}_i^{\mathcal{P}})\}_{i=1}^{N_{\mathcal{P}}}$, where $N_{\mathcal{P}}$ represents the number of protein atoms, $\mathbf{x}_i^{\mathcal{P}} \in \mathbb{R}^3$ represents the 3D coordinates of the protein atom, and $\mathbf{v}_i^{\mathcal{P}} \in \mathbb{R}^{N_f}$ represents protein atom features such as element types and amino acid types, with $N_f$ representing the number of such features. We aim to jointly optimize binding affinity and desired pharmacological properties (denoted as $\mathbb{Y}$), by generating prospective ligand molecules $\mathcal{M} = \{(\mathbf{x}_i^{\mathcal{M}}, \mathbf{v}_i^{\mathcal{M}})\}_{i=1}^{N_{\mathcal{M}}}$, for a given protein $\mathcal{P}$. Here $\mathbf{x}_i^{\mathcal{M}} \in \mathbb{R}^3$ and $\mathbf{v}_i^{\mathcal{M}} \in \mathbb{R}^K$ represents the atom coordinates and atom types of a ligand molecule, respectively, with $K$ representing the number of such features to represent atom types. The variable $N_{\mathcal{M}}$ signifies the number of atoms in the ligand molecule, which can be sampled during inference utilizing either an empirical distribution (Hoogeboom et al., 2022; Guan et al., 2023a;b) or predicted through a neural network (Liu et al., 2022). To simplify, in matrix representation the ligand molecule is denoted as $\mathbf{M} = [\mathbf{X}^{\mathcal{M}}, \mathbf{V}^{\mathcal{M}}]$, where $\mathbf{X}^{\mathcal{M}} \in \mathbb{R}^{N_{\mathcal{M}} \times 3}$ and $\mathbf{V}^{\mathcal{M}} \in \mathbb{R}^{N_{\mathcal{M}} \times K}$ and the protein is denoted as $\mathbf{P} = [\mathbf{X}^{\mathcal{P}}, \mathbf{V}^{\mathcal{P}}]$, where $\mathbf{X}^{\mathcal{P}} \in \mathbb{R}^{N_{\mathcal{P}} \times 3}$ and $\mathbf{V}^{\mathcal{P}} \in \mathbb{R}^{N_{\mathcal{P}} \times N_f}$. The foundational concepts integral to our research and the corresponding notations employed throughout this study are outlined in Appendix § C and Table 3 in the Appendix § A, respectively.

## 3. TAGMOL

In this work, we improve upon *Conditional Diffusion models* which utilizes protein pockets as a conditioning factor for generating ligand molecules (Guan et al., 2023a;b). The effectiveness of such models in practical scenarios can be hindered if the conditioning signal is overlooked or weakened, an issue that becomes more pronounced with datasets of inferior quality. Models that are exclusively trained to maximize the likelihood of protein-ligand complexes in such datasets naturally inherit the same quality issues. To counteract this, TAGMOL employs a strategic approach to explicitly integrate additional conditioning signals learned over a set of binding and desirable pharmacological properties $\mathbb{Y}$ (Recall § 2) during the model's denoising phase. In particular, for a property $\mathbf{y} \in \mathbb{Y}$, we train a regressor (or classifier as appropriate) $p_\phi(\mathbf{y}|\mathbf{M}_t, \mathbf{P}, t)$ on noisy molecule $\mathbf{M}_t$ and then use the gradients $\nabla_{\mathbf{X}_t^{\mathcal{M}}} p_\phi(\mathbf{y}|\mathbf{M}_t, \mathbf{P}, t)$ to guide the diffusion sampling process towards the desirable properties encoded in $\mathbf{y}$.

Figure 1 outlines the TAGMOL pipeline, which comprises two primary components. The first, *Guide Training*, detailed in § 3.1, introduces our *SE(3) Invariant* GNN architecture and its objectives, establishing the foundation of our

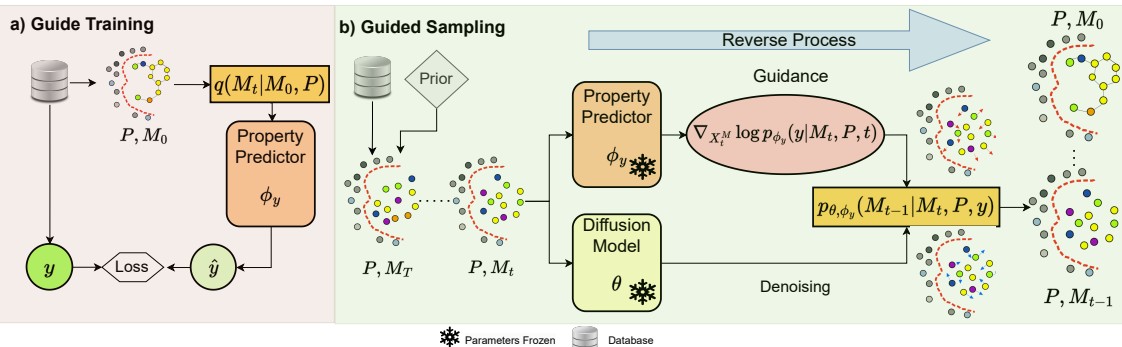

*Figure 1.* Overview of TAGMOL. (a) Training a property-oriented *guide* using existing data. (b) Utilizing the trained *guide* and *diffusion* model to steer the diffusion sampling process towards the optimal regions of the property of interest.

approach. The second component, discussed in § 3.2, is *Guided Sampling*, where we explain how a trained guide model effectively directs the diffusion sampling process toward specific regions of interest, emphasizing desired properties. Additionally, our study demonstrates the concurrent optimization of multiple properties, as elaborated in § 3.3.

### 3.1. Property Guide

We approach the input space as a 3D point cloud system where we build a $k$-nearest neighbors ($k$-NN) graph $\mathcal{G}$ by representing ligand and protein atoms as nodes, and each atom is connected to its $k$-nearest neighbors. For properties such as binding affinity, including protein atoms within the graph is critical; however, the inclusion of protein atoms can be omitted for properties solely dependent on the ligand. We parameterize our *guide* using an Invariant GNN $\phi_y$, which is trained to predict property $\mathbf{y} \in \mathbb{Y}$ given a noisy input $\mathbf{M}_t$, protein $\mathbf{P}$ and time $t$. Later in the denoising phase, the gradients of $\phi_y$ are used to direct exploration in regions of interest. Given the graph representation $\mathcal{G}$ at diffusion time step $t$, we define the GNN convolution layer as follows:

$$
\begin{aligned}
\mathbf{d}_{i,j} &= \mathcal{D}(\|\mathbf{x}_{i,t} - \mathbf{x}_{j,t}\|^2) \\
\mathbf{m}_{i,j} &= \phi_{\mathbf{y}}^m(\mathbf{h}_{i,t}^l, \mathbf{h}_{j,t}^l, \mathbf{d}_{i,j}, \mathbf{a}_{i,j}) \\
\mathbf{e}_{i,j} &= \phi_{\mathbf{y}}^e(\mathbf{m}_{i,j}) \\
\mathbf{m}_i &= \sum_{j \in \mathcal{N}(i)} \mathbf{e}_{i,j} \mathbf{m}_{i,j} \\
\mathbf{h}_{i,t}^{l+1} &= \phi_{\mathbf{y}}^h(\mathbf{h}_{i,t}^l, \mathbf{m}_i) \\
\mathbf{h}_{i,t}^0 &= \begin{cases} \phi_{\mathbf{y}}^p(\mathbf{v}_i^{\mathcal{P}}) & \text{if } i \in \mathcal{P} \\ \phi_{\mathbf{y}}^l([\mathbf{v}_{i,t}^{\mathcal{M}}, \tau]) & \text{if } i \in \mathcal{M} \end{cases}
\end{aligned}
\tag{1}
$$

where $\mathbf{h}_{i,t}^l \in \mathbb{R}^d$ represents the SE(3)-invariant hidden representation of protein and ligand atoms after $l$ layers; $\mathbf{x}_{i,t}, \mathbf{x}_{j,t} \in \mathbb{R}^3$ represents hidden representation of protein and ligand atoms. $\mathbf{d}_{i,j}$ represents the distance embedding and $\mathbf{a}_{i,j}$ represents the edge attributes in the graph. Invariance of $\mathbf{h}_{i,t}^l$ stems from the fact that we take the L2-distance between the atom representations. $\mathcal{N}(i)$ stands

for the set of neighbors for atom $i$. $\tau$ represents the time embedding to make the model aware of noise at time step $t$. $\phi_{\mathbf{y}}^m, \phi_{\mathbf{y}}^e, \phi_{\mathbf{y}}^h, \phi_{\mathbf{y}}^p$ and $\phi_{\mathbf{y}}^l$ are Multi-Layer Perceptrons (MLP) where as $\mathcal{D}$ is a distance encoder.

Once we get the final hidden states $\mathbf{h}_{i,t}^L$, we predict the final property using an MLP layer as:

$$
\hat{\mathbf{y}} = \phi_{\mathbf{y}}^f\left(\sum_{i \in \mathcal{M}} \mathbf{h}_{i,t}^L\right)
\tag{2}
$$

While equivariance is essential for generative models to maintain the consistency of the probability $p(\mathbf{M}_0|\mathbf{P})$ against *SE(3)* transformations in protein-ligand complexes, it is also vital that a scalar property predicted from *guide* remains independent of *SE(3)* transformations. Hence, the GNN in guide is SE(3) invariant.

The probability modeling is articulated through a normal distribution as:

$$
p_{\phi_y}(\mathbf{y}|\mathbf{M}_t, \mathbf{P}) = \mathcal{N}(\mathbf{y}, \phi_{\mathbf{y}}(\mathbf{M}_t, \mathbf{P}, t), \mathbf{I})
\tag{3}
$$

The training of these models is oriented toward minimizing the Negative log-likelihood ($NLL$).

$$
\begin{aligned}
NLL &= -\mathbb{E}_{p(\mathbf{P}, \mathbf{M}_{0:T})} \sum_{t=0}^{T} \log(p_{\phi_y}(\mathbf{y}|\mathbf{M}_t, \mathbf{P}, t)) \\
&= \mathbb{E}_{p(\mathbf{P}, \mathbf{M}_{0:T})} \sum_{t=0}^{T} \frac{(\mathbf{y} - \phi_{\mathbf{y}}(\mathbf{M}_t, \mathbf{P}, t))^2}{2}
\end{aligned}
\tag{4}
$$

Although we designed our guide for regression tasks, it is versatile enough to be adapted for classification tasks or other suitable applications.

### 3.2. Single Objective Guidance

Departing from the methodologies outlined in prior studies (Guan et al., 2023a;b), our generative model distinctively conditions on both the protein pocket and property to be guided. Remarkably, without retraining the diffusion model, we guide the diffusion sampling process by shifting coordinates.

As demonstrated in Dhariwal & Nichol (2021), efficiently sampling $\mathbf{M}_{t-1}$ for each denoising transition can be adequately achieved by:

$$p_{\theta,\phi_\mathbf{y}}(\mathbf{M}_{t-1}|\mathbf{M}_t,\mathbf{P},\mathbf{y}) = Z p_\theta(\mathbf{M}_{t-1}|\mathbf{M}_t,\mathbf{P}) \\ \cdot p_{\phi_\mathbf{y}}(\mathbf{y}|\mathbf{M}_{t-1},\mathbf{P},t-1) \quad (5)$$

where $Z$ is a normalizing constant. However, direct sampling from this distribution is intractable. We approximate the sampling process via perturbation to a Gaussian distribution as per prior work (Sohl-Dickstein et al., 2015; Dhariwal & Nichol, 2021). Thus, we use perturbation to $p_\theta(\mathbf{X}_{t-1}|\mathbf{M}_t,\mathbf{P})$ and sample $\mathbf{X}_{t-1}^{\mathcal{M}}$ as:

$$\mathbf{X}_{t-1}^{\mathcal{M}} \sim \mathcal{N}(\tilde{\mu}_\theta(\mathbf{M}_t,\mathbf{P},t) + s\tilde{\beta}\nabla_{\mathbf{X}_t^{\mathcal{M}}}\log p_{\phi_\mathbf{y}}(\mathbf{y}|\mathbf{M}_t,\mathbf{P},t),\tilde{\beta}\mathbf{I})$$
$$\mathbf{V}_{t-1}^{\mathcal{M}} \sim \mathcal{C}\left(\tilde{\boldsymbol{c}}_\theta\left(\mathbf{M}_t,\mathbf{P},t\right)\right) \quad (6)$$

In this context, $s$, denotes the guidance strength, playing a crucial role in prioritizing $\mathbf{M}_t$ with optimal property by sampling it from the updated distribution $\propto p_\theta(\mathbf{M}_{t-1}|\mathbf{M}_t,\mathbf{P})\left(p_{\phi_\mathbf{y}}(\mathbf{y}|\mathbf{M}_{t-1},\mathbf{P},t)\right)^s$. Fine-tuning of $s$ is essential during the optimization process while maintaining the effectiveness of denoising within the diffusion model.

The incorporation of discrete variables such as atom types $\mathbf{V}_t^{\mathcal{M}}$ in the diffusion process poses challenges in directly applying explicit guidance through gradients. However, in our approach, while we explicitly provide guidance for coordinates $\mathbf{X}_t^{\mathcal{M}}$, the guidance for discrete variables operates implicitly. More specifically, at diffusion time step $t$, the denoised ligand atom types $\mathbf{V}_{t-1}^{\mathcal{M}}$ are influenced by both $\mathbf{X}_t^{\mathcal{M}}$ and $\mathbf{V}_t^{\mathcal{M}}$. Therefore, by providing guidance for $\mathbf{X}_t^{\mathcal{M}}$, the generative model is encouraged to denoise $\mathbf{V}_{t-1}^{\mathcal{M}}$ for the optimized $\mathbf{X}_t^{\mathcal{M}}$, effectively making the guidance implicit for the atom types.

### 3.3. Multi Objective Guidance

We now generalize our objective to holistically enhance various desired properties, collectively denoted as $\mathbb{Y}$. At each denoising step $t$, we condition on $\mathbb{Y}$ and sample $\mathbf{M}_{t-1}$ according to the probability distribution as :

$$p_{\theta,\phi_\mathbb{Y}}(\mathbf{M}_{t-1}|\mathbf{M}_t,\mathbf{P},\mathbb{Y}) = Z p_\theta(\mathbf{M}_{t-1}|\mathbf{M}_t,\mathbf{P}) \\ \cdot p_{\phi_\mathbb{Y}}(\mathbb{Y}|\mathbf{M}_{t-1},\mathbf{P},t) \quad (7)$$

Assuming all the properties $\mathbf{y} \in \mathbb{Y}$ are conditionally independent given $\mathbf{M}_{t-1}$ and $\mathbf{P}$, Eq. 7 is factorized as follows:

$$p_{\theta,\phi_\mathbb{Y}}(\mathbf{M}_{t-1}|\mathbf{M}_t,\mathbf{P},\mathbb{Y}) = Z p_\theta(\mathbf{M}_{t-1}|\mathbf{M}_t,\mathbf{P}) \\ \cdot \prod_{\mathbf{y}\in\mathbb{Y}} p_{\phi_\mathbf{y}}(\mathbf{y}|\mathbf{M}_{t-1},\mathbf{P},t) \quad (8)$$

Thus we train a set of models $\phi_\mathbb{Y} = \{\phi_\mathbf{y} : \forall \mathbf{y} \in \mathbb{Y}\}$ independently and sample $\mathbf{M}_{t-1}$ using a similar approximated posterior distribution in Equation 6 as:

$$\mathbf{X}_{t-1}^{\mathcal{M}} \sim \mathcal{N}(\tilde{\mu}_\theta(\mathbf{M}_t,\mathbf{P},t) + \delta, \tilde{\beta}\mathbf{I}) \\ \mathbf{V}_{t-1}^{\mathcal{M}} \sim \mathcal{C}\left(\tilde{\boldsymbol{c}}_\theta\left(\mathbf{M}_t,\mathbf{P},t\right)\right) \quad (9)$$

where

$$\delta = \sum_{\mathbf{y}\in\mathbb{Y}} s_\mathbf{y}\tilde{\beta}\nabla_{\mathbf{X}_t^{\mathcal{M}}}\log p_{\phi_\mathbf{y}}(\mathbf{y}|\mathbf{M}_t,\mathbf{P},t) \quad (10)$$

and $s_\mathbf{y}$ represents the guidance strength for property $\mathbf{y} \in \mathbb{Y}$.

## 4. Experimental Setup

**Datasets** TAGMOL is trained and evaluated on the Cross-Docked2020 dataset (Francoeur et al., 2020), consistent with the approaches outlined in Luo et al. (2021); Peng et al. (2022); Guan et al. (2023a;b). The training set includes roughly 100,000 protein-ligand pairs, where the root-mean-square deviation (RMSD) between the docked pose and the ground truth is under 1Å, and the protein sequences exhibit less than 30% identity. For inference, we generate 100 molecules for each of the 100 proteins in the test set.

**Baselines** We benchmark against state-of-the-art baselines in structure-based drug design (SBDD). This includes liGAN (Ragoza et al., 2022), which leverages a conditional variational autoencoder (CVAE) trained on a grid representation of atomic densities in protein-ligand structures. Additionally, we consider AR (Luo et al., 2021) and Pocket2Mol (Peng et al., 2022), both GNN-based methods that employ autoregressive frameworks to generate 3D molecular atoms by conditioning on the protein pocket and previously generated atoms. Furthermore, we extend to the recent diffusion-based approaches, TargetDiff (Guan et al., 2023a) and DecompDiff (Guan et al., 2023b), which non-autoregressively generate atom coordinates and types. DecompDiff enhances TargetDiff with bond considerations and decomposed priors for ligand arms and scaffolds. For a comprehensive overview of prior works, refer to Appendix B.

**Metrics** In line with previous works, we assess the molecular properties of the generated compounds using the following metrics: (i) QED, which estimates drug-likeness (Bickerton et al., 2012); (ii) SA, an indicator of synthetic accessibility, normalized between 0 and 1 (Ertl & Schuffenhauer, 2009); and (iii) Diversity, determined by calculating the average pairwise Tanimoto distances between ligands (Bajusz et al., 2015; Tanimoto, 1958). To analyze metrics related to the binding affinity with the target, we employ AutoDock Vina (Eberhardt et al., 2021). The metrics include: (i) Vina Score, directly evaluating binding affinity based on the 3D molecular poses generated; (ii) Vina Min, which estimates affinity after local structure minimization; (iii) Vina Dock, which incorporates a re-docking process

*Table 1.* Comparison of various properties between reference molecules and those generated by our model and other baselines. (↑) / (↓) indicates whether a larger or smaller number is preferable. The first and second-place results are emphasized with bold and underlined text, respectively. Refer to Appendices F.5 and F.6 for more evaluations.

| Methods | Vina Score (↓) | | Vina Min (↓) | | Vina Dock (↓) | | High Affinity (↑) | | QED (↑) | | SA (↑) | | Diversity (↑) | | Hit(↑) |
| | Avg. | Med. | Avg. | Med. | Avg. | Med. | Avg. | Med. | Avg. | Med. | Avg. | Med. | Avg. | Med. | Rate % |
|---|---|---|---|---|---|---|---|---|---|---|---|---|---|---|---|
| Reference | -6.36 | -6.46 | -6.71 | -6.49 | -7.45 | -7.26 | - | - | 0.48 | 0.47 | 0.73 | 0.74 | - | - | 21 |
| liGAN | - | - | - | - | -6.33 | -6.20 | 21.1% | 11.1% | 0.39 | 0.39 | 0.59 | 0.57 | 0.66 | 0.67 | 13.2 |
| AR | -5.75 | -5.64 | -6.18 | -5.88 | -6.75 | -6.62 | 37.9% | 31.0% | 0.51 | 0.50 | 0.63 | 0.63 | 0.70 | 0.70 | 12.9 |
| Pocket2Mol | -5.14 | -4.70 | -6.42 | -5.82 | -7.15 | -6.79 | 48.4% | 51.0% | **0.56** | **0.57** | **0.74** | **0.75** | 0.69 | **0.71** | 24.3 |
| TargetDiff | -5.47 | -6.30 | -6.64 | -6.83 | -7.80 | -7.91 | 58.1% | 59.1% | 0.48 | 0.48 | 0.58 | 0.58 | **0.72** | **0.71** | 20.5 |
| DecompDiff | -4.85 | -6.03 | -6.76 | -7.09 | -8.48 | -8.50 | 64.8% | **78.6%** | 0.44 | 0.41 | 0.59 | 0.59 | 0.63 | 0.62 | 24.9 |
| TAGMOL | **-7.02** | **-7.77** | **-7.95** | **-8.07** | **-8.59** | **-8.69** | **69.8%** | 76.4% | 0.55 | 0.56 | 0.56 | 0.56 | 0.69 | 0.70 | **27.7** |

to gauge optimal binding affinity; and (iv) High Affinity, which measures the proportion of generated molecules exhibiting better binding than the reference molecule for each target protein. The percentage of molecules that satisfy the *hit criteria*—QED ≥ 0.4, SA ≥ 0.5, and Vina Dock ≤ -8.18 kcal/mol—is calculated. These thresholds align with those of marketed drugs (Eckmann et al., 2022), ensuring moderate biological activity. We refer to this as the *hit rate*.

**Generative Backbone**  We chose TargetDiff as our generative backbone model over DecompDiff. This decision was based on TargetDiff's self-contained nature, in contrast to DecompDiff, which relies heavily on external computational tools. Specifically, DecompDiff utilizes AlphaSpace2 (Katigbak et al., 2020) for extracting subpockets, which are potential protein binding sites. In fact, our study demonstrates how our guidance mechanism can effectively replace the computational tools employed in DecompDiff.

**Guide Training**  We train *guide* models for various properties employing an Invariant Graph Neural Network (GNN) architecture, as detailed in § 3.1. For QED and SA, which are target-independent properties, protein atoms are excluded during the $k$-Nearest Neighbors ($k$-NN) graph construction. To ensure a fair comparison with other baseline methods, we utilize the same CrossDocked2020 dataset and identical data splits when training the *guide* models. For training our binding affinity guide, we used Autodock Vina scores from the CrossDocked2020 dataset, and for QED and SA, we calculated scores using the RDKit package (Landrum, 2013). To effectively guide the denoising phase, we train our guide using the same noise as our backbone model, TargetDiff, which applies Gaussian noise to coordinates and uniform noise to categories. Additional training and evaluation details of *guide* can be found in Appendix E.

**Guide Strengths**  For the single objective guidance, the optimal guide strength $s_{\mathbf{y}}^{opt}$ for each property $\mathbf{y}$ is identified through a grid search on small set of 4 targets (see Appendix F.1 for more details). The configuration that delivers the best value for the intended property in the generated

molecules is selected. For multi-objective guidance, the optimal guide strength values $s_{\mathbf{y}}^{opt}$ are recalibrated using a set of weights $w_{\mathbf{y}}$, where $\sum_{\mathbf{y} \in \mathbb{Y}} w_{\mathbf{y}} = 1$. These modified guide strengths, $w_{\mathbf{y}} s_{\mathbf{y}}^{opt}$, are subsequently utilized to steer property optimization as described in Equation 9. In our approach, where all properties are considered equally important, we assigned equal weights to them.

## 5. Results

TAGMOL outperforms all baselines, including the reference molecules, in binding-related metrics and *hit rate* (see Table 1). This achievement is especially noteworthy when compared to the state-of-the-art DecompDiff model, which relies on external computation for informed priors in the denoising process. The success of our model, attained without requiring extra data, underscores the importance of effectively learning useful signals from the existing training set and skillfully guiding the diffusion denoising phase. Figure 2 presents examples of ligand molecules generated by our model, featuring valid structures and reasonable binding poses to the target.

**Binding scores**  Our TAGMOL excels in the Vina Score, achieving a 22% improvement over state of the art (AR). This highlights its proficiency not only in generating molecules that bind effectively with proteins but also in producing high-affinity poses. Further, 69.8% of the molecules generated by our model exhibit superior binding affinity compared to reference molecules, surpassing all other models.

**Molecular properties**  TAGMOL shows remarkable performance, surpassing diffusion-based models in QED by 14.6% and 22.2% compared to TargetDiff and DecompDiff, respectively, while maintaining similar diversity levels. Although there's a slight decrease in SA, the metric remains within a reasonable range, indicating satisfactory synthesizability. A detailed discussion of the challenges in optimizing SA is provided in Appendix F.7. Appendix F.4 presents evidence of statistically significant changes in guided properties

*Table 2.* Ablation analysis assessing the properties of generated molecules under different property guidance scenarios. The first and second-place results are emphasized with bold and underlined text, respectively.

| Methods | Vina Score (↓) | | Vina Min (↓) | | Vina Dock (↓) | | QED (↑) | | SA (↑) | | Hit (↑) |
| --- | --- | --- | --- | --- | --- | --- | --- | --- | --- | --- | --- |
| | Avg. | Med. | Avg. | Med. | Avg. | Med. | Avg. | Med. | Avg. | Med. | Rate |
| backbone | -5.47 | -6.30 | -6.64 | -6.83 | -7.80 | -7.91 | 0.48 | 0.48 | 0.58 | 0.58 | 20.5 |
| backbone + BA Opt | **-7.35** | **-8.18** | **-8.38** | **-8.46** | **-9.04** | **-9.04** | 0.49 | 0.50 | 0.53 | 0.53 | 22.6 |
| backbone + QED Opt | -5.48 | -6.46 | -6.77 | -6.93 | -7.93 | -8.06 | **0.56** | **0.57** | 0.58 | 0.58 | 24.5 |
| backbone + SA Opt | -5.22 | -6.03 | -6.40 | -6.57 | -7.53 | -7.73 | 0.47 | 0.48 | **0.61** | **0.60** | 19.4 |
| TAGMOL | -7.02 | -7.77 | -7.95 | -8.07 | -8.59 | -8.69 | 0.55 | 0.56 | 0.56 | 0.56 | **27.7** |

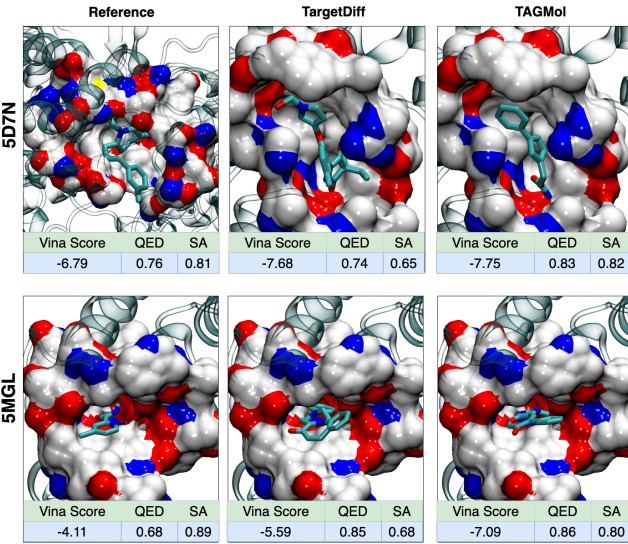

*Figure 2.* Visualization of reference molecules (left), alongside molecules generated by our backbone, TargetDiff (middle), and TAGMOL (right), for two targets: 5D7N and 5MGL.

upon incorporation of guidance.

**Ablation**   In Table 2, we conduct an ablation study on our property guidance mechanism for multiple properties separately and all of them in tandem, demonstrating its effectiveness across various aspects. We observe superior performance in each property when we provide respective guidance, showcasing the robustness of our guidance mechanism. We also observed that when we provide guidance for one property, it does not result in a substantial deterioration in other properties. This highlights the collaborative mechanism between the *denoising* model and the *guide* model, emphasizing their combined power. Notably, a substantial enhancement is observed when we guide for binding affinity. This improvement is likely due to our approach of calculating binding affinity through docking software (Trott & Olson, 2010), which primarily depends on atomic distances. Our guide model's architecture, with its inherent inductive bias in modeling atomic distances, effectively facilitates the explicit guidance for coordinate generation, contribut-

ing to this advancement. Consequently, improvements in QED and SA scores are comparatively less pronounced, as these properties exhibit a relatively lower dependence on molecule's geometric configuration. TAGMOL's focus on optimizing multiple properties, achieves a state-of-the-art *hit rate*, surpassing the backbone model—TargetDiff—by a considerable margin. This is evidenced by the higher fraction of molecules generated by our method that meet the *hit criteria*. These collective results underscore the formidable capabilities of TAGMOL in SBDD. Table 6 in Appendix, presents a more comprehensive study in evaluating the effectiveness of our guides when offering guidance for subsets of properties.

## 6. Conclusions

In this work, we redefined the problem of drug generative modeling by extending our focus beyond merely optimizing binding activity. We emphasize the importance of optimizing additional properties and integrating these considerations into our problem formulation, even when training data is sparse. Secondly, we developed a optimized generative method, TAGMOL, which effectively combines two key components: an *SE(3) equivariant graph diffusion model* to accurately replicate the geometries found in the training set and a multi-objective guidance driven by *SE(3) invariant GNN models*. This innovative combination enables our model to efficiently navigate the diffusion sampling space, focusing on properties of interest through the use of gradients. Lastly, our rigorous empirical evaluation demonstrates that TAGMOL notably enhances metric performance, balancing the optimization of binding affinity with critical pharmacological properties such as QED and SA.

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

# A. Notation Summary

Table 3. Notations

| Notation | Explanation | Domain |
|---|---|---|
| $N_{\mathcal{P}}$ | Number of protein atoms | $\in \mathbb{R}^1$ |
| $N_f$ | Feature dimension of protein atom | $\in \mathbb{R}^1$ |
| $\mathbf{x}_i^{\mathcal{P}}$ | $i$-th protein atom coordinate | $\in \mathbb{R}^3$ |
| $\mathbf{v}_i^{\mathcal{P}}$ | $i$-th protein atom feature | $\in \mathbb{R}^{N_f}$ |
| $N_{\mathcal{M}}$ | Number of ligand atoms | $\in \mathbb{R}^1$ |
| $K$ | Number of ligand atom types | $\in \mathbb{R}^1$ |
| $\mathbf{x}_i^{\mathcal{M}}$ | $i$-th ligand atom coordinate | $\in \mathbb{R}^3$ |
| $\mathbf{x}_{i,t}^{\mathcal{M}}$ | $i$-th ligand atom coordinate at diffusion time step $t$ | $\in \mathbb{R}^3$ |
| $\mathbf{v}_i^{\mathcal{M}}$ | $i$-th ligand atom *one-hot* representation | $\in \mathbb{R}^K$ |
| $\mathbf{v}_{i,t}^{\mathcal{M}}$ | $i$-th ligand atom *one-hot* representation at diffusion time step $t$ | $\in \mathbb{R}^K$ |
| $\mathbf{X}^{\mathcal{M}}$ | Matrix representation of all ligand atom coordinates | $\in \mathbb{R}^{N_{\mathcal{M}} \times 3}$ |
| $\mathbf{V}^{\mathcal{M}}$ | Matrix representation of all ligand atom types | $\in \mathbb{R}^{N_{\mathcal{M}} \times K}$ |
| $\mathbf{X}^{\mathcal{P}}$ | Matrix representation of all protein atom coordinates | $\in \mathbb{R}^{N_{\mathcal{P}} \times 3}$ |
| $\mathbf{V}^{\mathcal{P}}$ | Matrix representation of all protein atom features | $\in \mathbb{R}^{N_{\mathcal{P}} \times N_f}$ |
| $\beta_t$ | Variance Schedule for the diffusion model | $\in \mathbb{R}^1$ |
| $q$ | Diffusion noising transition function | - |
| $\theta$ | Parameters of generative model | - |
| $p_\theta$ | Denoising diffusion transition function | - |
| $\mathbb{Y}$ | Set of properties to be optimized during generation | - |
| $\mathbf{y}$ | A property $\mathbf{y} \in \mathbb{Y}$ | - |
| $\phi_{\mathbf{y}}$ | Parameters of guide corresponding to $\mathbf{y}$ | - |
| $\phi_{\mathbb{Y}}$ | Set of guide parameters corresponding to $\forall \mathbf{y} \in \mathbb{Y}$ | - |
| $s_{\mathbf{y}}$ | Guide strength for property $\mathbf{y}$ | $\in \mathbb{R}^1$ |
| $s_{\mathbf{y}}^{opt}$ | Optimal Guide strength for property $\mathbf{y}$ | $\in \mathbb{R}^1$ |

# B. Related Work

**Neural Drug Design** Neural models have significantly changed the landscape for drug design that was previously dominated by computational techniques such as MD simulations and Docking (Alonso et al., 2006). It has gone through multiple paradigm shifts in the recent past - ranging from using only 1D representations (SMILES) (Bjerrum & Threlfall, 2017; Gómez-Bombarelli et al., 2018) to 2D representations (molecular graphs) (Liu et al., 2018; Zhou et al., 2018) to 3D representations (Skalic et al., 2019; Ragoza et al., 2020) (molecular coordinates). Recent works have also showcased the importance of using target-aware models for practical drug design. Advancements in target-aware drug design, particularly in the realm of text and graph-based generative methods (Eckmann et al., 2022; Stanton et al., 2022; Chenthamarakshan et al., 2020; Lee et al., 2023), have made significant strides. However, these methods often generate ligands without considering the 3D structure of the target protein. TargetDiff (Guan et al., 2023a) and DecompDiff (Guan et al., 2023b) are two diffusion-based models that consider the 3D structure of the target protein. DecompDiff extends TargetDiff by decomposing the task into generating the arms and scaffold while explicitly considering bonds between the atoms during the diffusion process which is ignored by TargetDiff. Fragment-based generation methods such as (Ghorbani et al., 2023) allow for controlled generation, ensuring that only certain types of molecular fragments are present in the generated ligand. But they still fall behind the performance of diffusion-based approaches in terms of their binding affinity.

**Molecular Property Prediction** Predicting the properties of molecules through physio-chemical experiments is an expensive and time-consuming solution, that is unsuitable for getting intermediate feedback on AI-designed molecules. Using neural models has allowed high-quality prediction of various molecular properties in an automated fashion. Neural methods use different molecular representations to predict molecules, including SMILES representation with pre-trained transformers (Liu et al., 2023) or molecular structures (Yang et al., 2019; Zhou et al., 2023) for predicting the properties of interest.

**Molecular Property Optimization** Prior methods for property optimization have focused on using Reinforcement learning to generate molecules with desired properties (You et al., 2018; Zhou et al., 2019; Zhavoronkov et al., 2019; Jeon & Kim, 2020; Olivecrona et al., 2017). However, RL methods are often computationally expensive and challenging to optimize due to the vast search space. LIMO (Eckmann et al., 2022) uses Variational Auto Encoders (VAEs) and gradient gudiance on MLP-based property predictors but does not consider the 3D structure of the target protein or the generated ligand. (Lee et al., 2023) introduced a novel graph-based diffusion model for out-of-distribution (OOD) generation, enhancing explorability through OOD-controlled reverse-time diffusion and property-guided sampling, albeit focusing solely on 2D molecular representations, unlike the target-aware 3D structure generation with multi-objective guidance featured in our work.

## C. Diffusion models for Target-Aware Generation

As delineated in previous works on diffusion-based target-aware molecular generations (Guan et al., 2023a;b), the process involves two phases: *noise injection*, also termed as forward diffusion, and *denoising* (backward diffusion).

**Noise injection:** This phase involves a gradual injection of Gaussian noise for co-ordinates and uniform noise for categorical data through a Markov chain. This noise addition is uniquely applied to the ligand molecule, excluding the protein in the diffusion process. In this context, the atom positions and types of the ligand molecule at time step $t$ are represented as $\mathbf{X}_t^{\mathcal{M}}$ and $\mathbf{V}_t^{\mathcal{M}}$ respectively. The diffusion forward transition is defined by the following equations:

$$q\left(\mathbf{M}_t|\mathbf{M}_{t-1}, \mathbf{P}\right) = \mathcal{N}\left(\mathbf{X}_t^{\mathcal{M}}; \sqrt{1-\beta_t}\mathbf{X}_{t-1}^{\mathcal{M}}, \beta_t\mathbf{I}\right)$$
$$\cdot \mathcal{C}\left(\mathbf{V}_t^{\mathcal{M}}|\left(1-\beta_t\right)\mathbf{V}_{t-1}^{\mathcal{M}} + \beta_t/K\right) \tag{11}$$

$$q\left(\mathbf{M}_t|\mathbf{M}_0, \mathbf{P}\right) = \mathcal{N}\left(\mathbf{X}_t^{\mathcal{M}}; \sqrt{\bar{\alpha}_t}\mathbf{X}_0^{\mathcal{M}}, \left(1-\bar{\alpha}_t\right)\mathbf{I}\right)$$
$$\cdot \mathcal{C}\left(\mathbf{V}_t^{\mathcal{M}}|\bar{\alpha}_t\mathbf{V}_0^{\mathcal{M}} + \left(1-\bar{\alpha}_t\right)/K\right) \tag{12}$$

where $\mathcal{N}$ and $\mathcal{C}$ denotes the normal and categorical distribution respectively while $\beta_1, ..., \beta_T$ represents the variance schedules. The corresponding posteriors are analytically derived as follows:

$$q\left(\mathbf{M}_{t-1}|\mathbf{M}_0, \mathbf{M}_t, \mathbf{P}\right) = \mathcal{N}\left(\mathbf{X}_{t-1}^{\mathcal{M}}; \tilde{\mu}_t\left(\mathbf{X}_t^{\mathcal{M}}, \mathbf{X}_0^{\mathcal{M}}\right), \tilde{\beta}_t\mathbf{I}\right)$$
$$\cdot \mathcal{C}\left(\mathbf{V}_{t-1}^{\mathcal{M}}|\tilde{c}_t\left(\mathbf{V}_t^{\mathcal{M}}, \mathbf{V}_0^{\mathcal{M}}\right)\right) \tag{13}$$

where,

$\tilde{\mu}_t\left(\mathbf{X}_t^{\mathcal{M}}, \mathbf{X}_0^{\mathcal{M}}\right) = \frac{\sqrt{\bar{\alpha}_{t-1}}\beta_t}{1-\bar{\alpha}_t}\mathbf{X}_0^{\mathcal{M}} + \frac{\sqrt{\alpha_t}\left(1-\bar{\alpha}_{t-1}\right)}{1-\bar{\alpha}_t}\mathbf{X}_t^{\mathcal{M}}, \tilde{\beta}_t = \frac{1-\bar{\alpha}_{t-1}}{1-\bar{\alpha}_t}\beta_t, \alpha_t = 1-\beta_t, \bar{\alpha}_t = \Pi_{s=1}^t\alpha_s,$
$\tilde{c}_t\left(\mathbf{V}_t^{\mathcal{M}}, \mathbf{V}_0^{\mathcal{M}}\right) = \mathbf{c}^\star / \sum_{k=1}^K c_k^\star$, and $\mathbf{c}^\star\left(\mathbf{V}_t^{\mathcal{M}}, \mathbf{V}_0^{\mathcal{M}}\right) = \left[\alpha_t\mathbf{V}_t^{\mathcal{M}} + \left(1-\alpha_t\right)/K\right] \odot \left[\bar{\alpha}_{t-1}\mathbf{V}_0^{\mathcal{M}} + \left(1-\bar{\alpha}_{t-1}\right)/K\right].$

In practical applications, it is recognized that the schedules $\beta_t$ for coordinates and categories may differ. However, for the sake of simplicity in this context, they are uniformly represented.

**Denoising phase:** In the generative process, a neural network parameterized by $\theta$ learns to recover $\mathbf{M}_0$ by iteratively denoising $\mathbf{M}_T$. During reverse process, $\mathbf{M}_0$ is approximated using $\theta$ and $\mathbf{M}_{t-1}$ by predicting $\widehat{\mathbf{M}}_{0|t} = [\widehat{\mathbf{X}}_{0|t}^{\mathcal{M}}, \widehat{\mathbf{V}}_{0|t}^{\mathcal{M}}]$ at time step $t$ and $\mathbf{M}_{t-1}$ is sampled as follows:

$$p_\theta\left(\mathbf{M}_{t-1}|\mathbf{M}_t, \mathbf{P}\right) = q\left(\mathbf{M}_{t-1}|\widehat{\mathbf{M}}_{0|t}, \mathbf{M}_t, \mathbf{P}\right)$$
$$= \mathcal{N}\left(\mathbf{X}_{t-1}^{\mathcal{M}}; \tilde{\mu}_\theta\left(\mathbf{M}_t, \mathbf{P}, t\right), \tilde{\beta}_t\mathbf{I}\right)$$
$$\cdot \mathcal{C}\left(\mathbf{V}_{t-1}^{\mathcal{M}}|\tilde{c}_\theta\left(\mathbf{M}_t, \mathbf{P}, t\right)\right) \tag{14}$$
$$= \mathcal{N}\left(\mathbf{X}_{t-1}^{\mathcal{M}}; \tilde{\mu}_t\left(\mathbf{X}_t^{\mathcal{M}}, \hat{\mathbf{X}}_{0|t}^{\mathcal{M}}\right), \tilde{\beta}_t\mathbf{I}\right)$$
$$\cdot \mathcal{C}\left(\mathbf{V}_{t-1}^{\mathcal{M}}|\tilde{c}_t\left(\mathbf{V}_t^{\mathcal{M}}, \hat{\mathbf{V}}_{0|t}^{\mathcal{M}}\right)\right)$$

**Training:** In line with the principles outlined in the variational auto-encoder (Kingma & Welling, 2013), the model undergoes training through the optimization of variational bound on the negative log-likelihood. Given that both $q\left(\mathbf{X}_{t-1}|\mathbf{M}_0, \mathbf{M}_t, \mathbf{P}\right)$ and $p_\theta\left(\mathbf{X}_{t-1}|\mathbf{M}_t, \mathbf{P}\right)$ are Gaussian distributions, the Kullback-Leibler (KL) divergence for the atom coordinates is expressed in a closed-form equation:

$$L_{t-1}^{(x)} = \frac{1}{2\tilde{\beta}_t} \|\tilde{\mu}_t \left( \mathbf{X}_t^{\mathcal{M}}, \mathbf{X}_0^{\mathcal{M}} \right) - \tilde{\mu}_\theta \left( [\mathbf{X}_t^{\mathcal{M}}, \mathbf{V}_t^{\mathcal{M}}], \mathbf{P}, t \right) \|^2 + C$$

$$= \gamma_t \|\mathbf{X}_0^{\mathcal{M}} - \widehat{\mathbf{X}}_{0|t}^{\mathcal{M}}\|^2 + C \tag{15}$$

where $\gamma_t = \frac{\bar{\alpha}_{t-1}\beta_t^2}{2\sigma_t^2(1-\bar{\alpha}_t)^2}$ and $C$ is a constant. As recommended by (Ho et al., 2020) and (Guan et al., 2023a), training the model using an unweighted Mean Squared Error (MSE) loss, by setting $\gamma_t = 1$, leads to enhanced performance. Regarding the atom-type loss, the KL divergence of categorical distributions is computed in the following manner:

$$L_{t-1}^{(v)} = \sum_{k=1}^{K} \boldsymbol{c}\left(\mathbf{V}_t^{\mathcal{M}}, \mathbf{V}_0^{\mathcal{M}}\right)_k \log \frac{\boldsymbol{c}\left(\mathbf{V}_t^{\mathcal{M}}, \mathbf{V}_0^{\mathcal{M}}\right)_k}{\boldsymbol{c}\left(\mathbf{V}_t^{\mathcal{M}}, \widehat{\mathbf{V}}_{0|t}^{\mathcal{M}}\right)_k} \tag{16}$$

The final loss is a weighted sum of atom coordinate loss and atom type loss, which is expressed as $L = L_{t-1}^{(x)} + \lambda L_{t-1}^{(v)}$.

## D. Property Distribution in Training Data

Figure 3 illustrates the distribution of QED, SA, and Vina Scores within the training split. From the figure, it is evident that a substantial proportion of molecules exhibit low binding and fall below the hit criteria for QED.

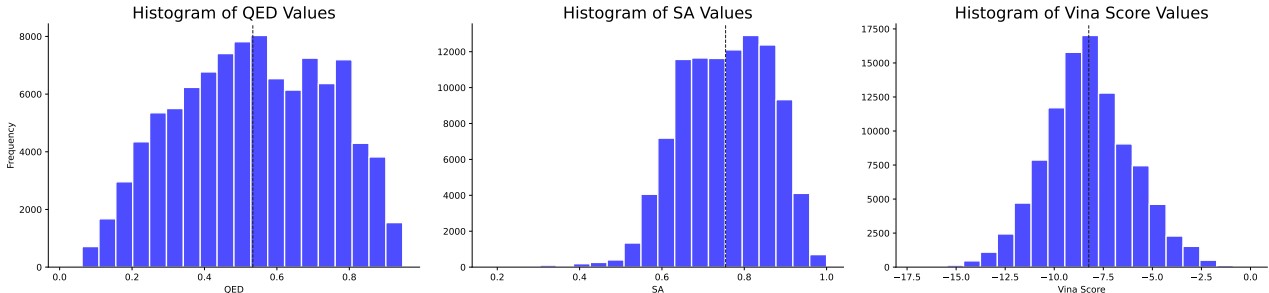

*Figure 3.* Distribution of QED, SA, and Vina Score properties in the training split, with the mean values represented by dotted lines

## E. Training and Evaluation of Guide

### E.1. Training

As detailed in § 3.1, we employed a 9-layer Invariant Graph Neural Network (GNN) as the foundational model for constructing *guides* across various properties. Notably, for QED and SA — properties that are not dependent on the target — we adapted the $k$-Nearest Neighbors ($k$-NN) graph construction methodology, which omits protein atoms into consideration. The $k$ value was specifically tailored to each property: set at 6 for both QED and SA and increased to 32 for the binding affinity property predictor to optimize the model's performance in different contexts. For training our *binding affinity guide*, we used Autodock Vina scores from the CrossDocked2020 dataset (Eberhardt et al., 2021; Francoeur et al., 2020), and for QED and SA, we calculated scores using the RDKit package (Landrum, 2013). To effectively guide the denoising phase, we train our guide using the same noise as our backbone model, TargetDiff, which applies Gaussian noise to coordinates and uniform noise to categories.

The models are trained with a batch size of 4 to minimize the Root Mean Square Error (RMSE) loss, utilizing a gradient descent algorithm. The initial learning rates were set at 0.001 for the *binding affinity* and *QED guides*, and at 5e-4 for the *SA guide*. Additionally, to maintain the stability of the training process, the gradient norms were clipped at a maximum value of 8. The learning rate underwent exponential decay by a factor of 0.6, with a lower bound set at 1e-5. This decay was triggered in the absence of any improvement in the validation loss across 10 consecutive evaluations. Evaluations were conducted every 2000 training steps, and convergence of all models was achieved within 200,000 steps. All experiments were conducted using the NVIDIA A100 GPU.

## E.2. Evaluation

The performance of each guide model was evaluated for their respective property predictions using the same test dataset. As outlined in § 4, we employed the same split to evaluate our *binding affinity guide* model. While the train-test split ensures the uniqueness of the protein-ligand complexes between the sets, 34 ligands are common to both, leading to potential train-test leakage. Consequently, for *QED* and *SA* guides that depend solely on ligands, we have excluded these 34 overlapping ligands from the test split (No overlaps). Table 4 presents the performance evaluation of each *guide* model, indicating high $R^2$ values on the original test dataset. After removing the overlapping ligands from the test set, the performance metrics remained consistent, confirming the model's effective generalization to unseen ligands. While we report the results on fair test split, it's crucial to note that, diverging from the traditional machine learning training and testing process, we train the guide to direct the generated latent space of the diffusion model.

*Table 4.* Evaluation of the guide model on the original test set with unique protein-ligand complexes, and on a modified test set (No Overlaps) where 32 overlapping ligands have been removed.

| Target Property | Test Set (Original) | | Test Set (No Overlaps) | |
|---|---|---|---|---|
| | RMSE | $R^2$ | RMSE | $R^2$ |
| BA | 0.642 | 0.953 | 0.652 | 0.951 |
| QED | 0.067 | 0.892 | 0.079 | 0.850 |
| SA | 0.054 | 0.848 | 0.065 | 0.821 |

## F. Additonal Experiments and Results

### F.1. Search for optimal Guide Strength

To determine the optimal Guide Strength ($s$), a grid search was conducted over the set of values: $\{0, 1, 2, 5, 10, 20, 50\}$. Here, $s = 0$ represents the scenario without guidance, included for comparative purposes to gauge guide-induced enhancements. The generation process was limited to 50 molecules for a limited set of 4 targets, to accommodate computational constraints. Initially, for each property, we find the optimal guide strength $s_{\mathbf{y}}^{opt}$ over the grid. Once the optimal values for single objective are tuned, for multi-objective guidance, we obtained optimal $s$ values by recalibrating the $s_{\mathbf{y}}^{opt}$ with $w_{\mathbf{y}}$, as detailed in §-4.

Table 5 presents the results of our tuning across different guide strengths ($s$) mentioned in the grid. It is evident that non-zero $s$ values yield improved property metrics, underscoring the efficacy of guidance. For Binding Affinity (BA) guidance, optimal $s$ values emerge as 1 and 2, based on average and median Vina Scores, respectively. Considering both Vina Min and Vina Dock values, $s = 2$ is selected for superior outcomes. In the case of QED guidance, an $s$ value of 20 distinctly outperforms others in terms of both average and median QED scores. With SA guidance, although $s = 20$ and $s = 50$ appear to be optimal, the validity(Guan et al., 2023b) of generated molecules takes a low value of 67% and 46%, respectively. Consequently, $s = 5$ is chosen, providing the second-best SA values, with 84% of molecules being valid and only a slight decrease in average SA value by 0.02. In summary, the selected Guide Strengths are: $s_{ba}^{opt} = 2$, $s_{qed}^{opt} = 20$, and $s_{sa}^{opt} = 5$.

### F.2. Ablation on Single and Multi-Objective Guidance

To assess the impact of guidance on various combinations of properties, we conducted a detailed ablation study. Initially, we present results for single-objective guidance on different properties, then proceed to guidance for two properties at a time, and ultimately, guidance incorporating all properties. Our results, highlighted in Table 6, indicate that single-objective guidance maximizes improvement for the targeted property without any substantial deterioration in other properties. When applying guidance to two properties, the enhancements for each are moderate, illustrating a trade-off compared to separate guidance. This trade-off persists with guidance across all properties. However, given the multifaceted nature of drug discovery, where the goal is to produce molecules with high binding affinity and desired properties like QED and SA, our approach significantly enhances *hit rate* and most properties over no guidance. This underscores the effectiveness of guidance in navigating regions that are crucial in discovery.

*Table 5.* Search for optimal Guide Strength ($s$), conducted individually for each Guide Model (single-objective setting). To minimize computational demands, the generation is limited to 50 samples across 4 targets. Only the metrics of the property being guided are included. Symbols (↑) / (↓) denote preference for higher or lower values, respectively. Top results are highlighted in bold for first place and underlined for second. Italicized figures indicate instances where over 25% of molecules generated under the specified guide strength were not valid.

| | BA Guide | | | | | | QED Guide | | SA Guide | |
|---|---|---|---|---|---|---|---|---|---|---|
| | Vina Score (↓) | | Vina Min (↓) | | Vina Dock (↓) | | QED (↑) | | SA (↑) | |
| $s$ | Avg. | Med. | Avg. | Med. | Avg. | Med. | Avg. | Med. | Avg. | Med. |
| 0 | -5.37 | -6.17 | -6.31 | -6.83 | -7.65 | -7.67 | 0.51 | 0.54 | 0.59 | 0.59 |
| 1 | **-7.82** | -7.93 | -8.16 | -8.24 | -8.62 | -8.64 | 0.52 | 0.53 | 0.59 | 0.59 |
| 2 | -7.69 | **-8.08** | **-8.33** | **-8.40** | **-8.86** | **-8.89** | 0.52 | 0.53 | 0.60 | 0.59 |
| 5 | -7.55 | -7.74 | -8.02 | -8.04 | -8.67 | -8.65 | 0.55 | 0.58 | 0.62 | 0.62 |
| 10 | -7.11 | -7.09 | -7.53 | -7.33 | -8.42 | -8.34 | 0.54 | 0.54 | 0.60 | 0.59 |
| 20 | -6.85 | -6.69 | -7.03 | -6.87 | -8.16 | -8.14 | **0.58** | **0.60** | *0.64* | *0.63* |
| 50 | -6.11 | -5.96 | -5.68 | -5.91 | -7.73 | -7.44 | 0.55 | 0.55 | *0.64* | *0.63* |

*Table 6.* Extensive ablation analysis assessing the properties of generated molecules under different property guidance scenarios. The first and second-place results are emphasized with bold and underlined text, respectively.

| $w_{ba}$ | $w_{qed}$ | $w_{sa}$ | Vina Score (↓) | | Vina Min (↓) | | Vina Dock (↓) | | QED (↑) | | SA (↑) | | Hit (↑) |
|---|---|---|---|---|---|---|---|---|---|---|---|---|---|
| | | | Avg. | Med. | Avg. | Med. | Avg. | Med. | Avg. | Med. | Avg. | Med. | Rate |
| 0 | 0 | 0 | -5.47 | -6.30 | -6.64 | -6.83 | -7.80 | -7.91 | 0.48 | 0.48 | 0.58 | 0.58 | 20.5 |
| 1 | 0 | 0 | **-7.35** | **-8.18** | **-8.38** | **-8.46** | **-9.04** | **-9.04** | 0.49 | 0.5 | 0.53 | 0.53 | 22.6 |
| 0 | 1 | 0 | -5.48 | -6.46 | -6.77 | -6.93 | -7.93 | -8.06 | **0.56** | **0.57** | 0.58 | 0.58 | 24.5 |
| 0 | 0 | 1 | -5.22 | -6.03 | -6.4 | -6.57 | -7.53 | -7.73 | 0.47 | 0.48 | **0.61** | **0.6** | 19.4 |
| 0.5 | 0.5 | 0 | -7.11 | -7.96 | -8.13 | -8.21 | -8.82 | -8.85 | **0.56** | **0.57** | 0.55 | 0.55 | 26.1 |
| 0.5 | 0 | 0.5 | -7.2 | -7.95 | -8.16 | -8.26 | -8.83 | -8.84 | 0.5 | 0.51 | 0.55 | 0.55 | 24.7 |
| 0 | 0.5 | 0.5 | -5.43 | -6.34 | -6.63 | -6.86 | -7.85 | -7.97 | 0.55 | 0.56 | 0.59 | 0.59 | 24.1 |
| 0.34 | 0.33 | 0.33 | -7.02 | -7.77 | -7.95 | -8.07 | -8.59 | -8.69 | 0.55 | 0.56 | 0.56 | 0.56 | **27.7** |

## F.3. Multi-Objective Rationale

Figure 4 presents the distribution of molecular properties for 10,000 generated molecules (100 per target, across 100 targets). These molecules are generated using three methods: (i) the backbone model without any guidance; (ii) guidance using only two properties, leaving out the third one indicated on the x-axis; and (iii) guidance using all three properties. Optimizing all the properties helps improve the Vina Score and QED compared to the 'No Opt' baseline. Although it doesn't help improve the SA, it is still helpful, as excluding the SA property in optimization worsens the SA scores. Thus, it is useful to provide guidance for all three properties.

## F.4. Statistical Significance of Guidance

To demonstrate that the integration of guidance contributes to statistically significant changes in the results, we employ Paired $t$-test (for QED, SA, Vina Score, Vina Min and Vina Dock) and Chi-square test (for *hit rate*), comparing our model, TAGMOL, against the backbone model, TargetDiff. For the Paired $t$-test, we group the generated samples by target protein and compute the average scores for each property, yielding 100 pairs for comparison in our case. Our null hypothesis posits that there is no difference in the average values of a given property produced by the two models across various protein targets. The $p$-values for the guided properties QED, SA, and Vina Score were remarkably low, at 4.65E-33, 5.00E-13, and 7.05E-10, respectively. Similarly, for Vina Min and Vina Dock, the p-values were 2.89E-15 and 4.38E-06, respectively.

On the other hand, for the *hit rate*, we utilize the Chi-square test, since the $t$-test may not be the suitable test. Here, outcomes for a protein target are categorized as either "a greater number of molecules generated by TAGMOL satisfied the *hit criteria* compared to those by TargetDiff" or the reverse, with the null hypothesis being an equal likelihood of occurrence, implying

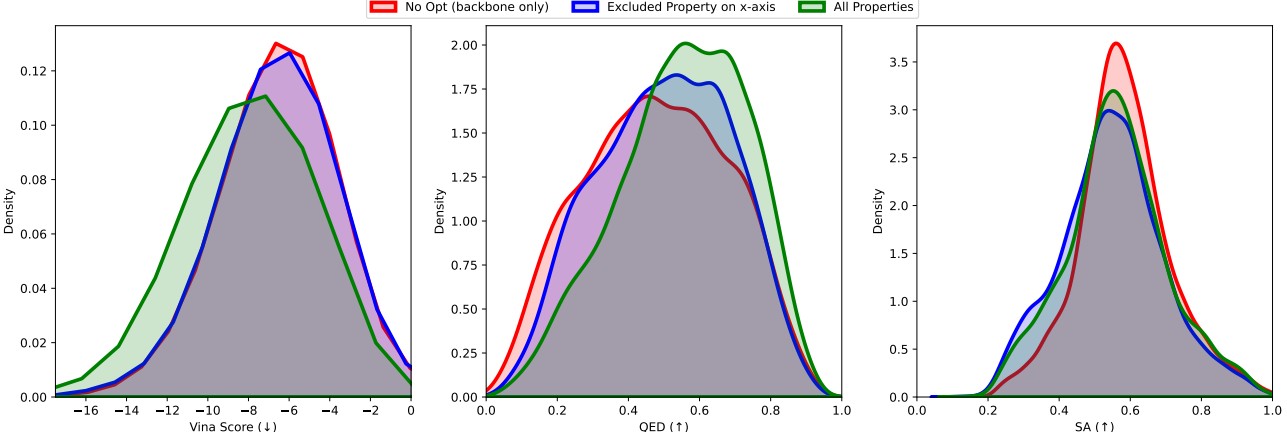

*Figure 4.* Distribution of molecular properties in molecules generated by the backbone model without any guidance (No Opt), when guided by two properties while excluding the one indicated on the x-axis (Excluded Property on x-axis), and when guided by all three properties (All Properties). (↓) denote properties where lower values are preferred, while (↑) indicate properties where higher values are desirable.

no difference between the models. The obtained $p$-value of 5.73E-07 further supports our findings.

Given that the $p$-values across all tests are substantially lower the than the conventional threshold of 0.05, we can confidently reject the null hypothesis, affirming that the guidance incorporation leads to highly statistically significant differences in the outcomes.

## F.5. Comparison of Bond Distance Distributions

*Table 7.* Jensen-Shannon Divergence comparing bond distance distributions between reference molecules and generated molecules. Lower values indicate better performance. '-' represents single bonds, '=' represents double bonds, and ':' represents aromatic bonds. The first and second-place results are emphasized with bold and underlined text, respectively.

| Bond | liGAN | AR | Pocket2 Mol | Target Diff | Decomp Diff | TAGMOL |
|------|-------|-----|-------------|-------------|-------------|--------|
| C-C  | 0.601 | 0.609 | 0.496 | **0.369** | 0.371 | 0.384 |
| C=C  | 0.665 | 0.620 | 0.561 | 0.505 | 0.539 | **0.501** |
| C-N  | 0.634 | 0.474 | 0.416 | 0.363 | **0.352** | 0.365 |
| C=N  | 0.749 | 0.635 | 0.629 | **0.550** | 0.592 | 0.559 |
| C-O  | 0.656 | 0.492 | 0.454 | 0.421 | **0.373** | 0.422 |
| C=O  | 0.661 | 0.558 | 0.516 | 0.461 | **0.381** | 0.430 |
| C:C  | 0.497 | 0.451 | 0.416 | 0.263 | **0.258** | 0.269 |
| C:N  | 0.638 | 0.551 | 0.487 | **0.235** | 0.273 | 0.252 |

We calculate the Jensen-Shannon divergences (JSD) to assess the differences in bond distance distributions between the reference molecules and the generated molecules (Guan et al., 2023a;b). As shown in Table 7, our method closely maintains the bond distribution in alignment with the backbone diffusion model, namely TargetDiff, surpassing the non-diffusion baselines. This outcome emphasizes the efficient synergy between our *guide* model and the *diffusion* model, markedly improving our ability to generate molecules with targeted properties while preserving molecular conformation.

## F.6. Benchmarking with PoseCheck

Expanding our evaluation, we utilized PoseCheck (Harris et al., 2023) to evaluate the 3D poses generated by the models. Our evaluation is focused on steric clashes and strain energy. Steric clashes quantify instances where the pairwise distance between a protein and ligand atom is below the sum of their van der Waals radii, with a clash tolerance of 0.5 Å. Meanwhile,

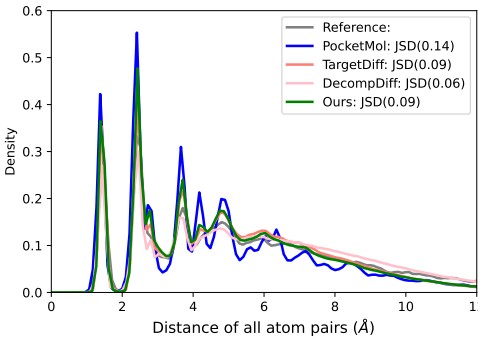

*Figure 5.* Comparison of distance distributions between all-atom distances of reference molecules in the test set (Reference) and distances in model-generated molecules. The Jensen-Shannon divergence (JSD) between these two distributions is reported.

strain energy represents the internal energy accumulated within a ligand due to conformational changes upon binding.

In Table 8 and 9, we present the mean values of steric clashes and the median values of strain energy for the generated molecules both before and after docking. In our discussion, we prioritize median values for strain energy as they offer greater representativeness in this context, especially given the presence of significant outliers.

TAGMOL demonstrates superior performance compared to diffusion-based models TargetDiff and DecompDiff in terms of clashes, highlighting the efficacy of our binding affinity guide. However, TAGMOL exhibits shortcomings concerning strain energy. As elucidated in PoseCheck, diffusion-based methodologies often yield elevated strain energies, a trend also observed in TAGMol due to minor errors in atom placement resulting from coordinate guidance. We hypothesize that integrating strain energy guidance during the generation phase could yield more stable molecules. We plan to investigate this in our future work.

*Table 8.* Evaluation of poses of generated molecules on biophysical benchmarks, PoseCheck. (↑) / (↓) indicates whether a larger or smaller value is preferable. The first-place results are emphasized in bold.

| Methods | Clashes (↓) | Strain Energy (↓) |
|---|---|---|
| TargetDiff | 9.19 | 1258.02 |
| DecompDiff | 12.53 | **539.89** |
| TAGMOL | **6.05** | 2143.03 |

*Table 9.* Evaluation of poses of generated molecules on biophysical benchmarks, PoseCheck, after redocking. (↑) / (↓) indicates whether a larger or smaller value is preferable. The first-place results are emphasized in bold.

| Methods | Clashes (↓) | Strain Energy (↓) |
|---|---|---|
| Reference | 3.5 | 118.17 |
| TargetDiff | 5.9 | 602.00 |
| DecompDiff | 5.75 | **441.61** |
| TAGMOL | **5.36** | 709.35 |

## F.7. Challenges in SA optimization

The SA score serves as a comprehensive metric for assessing synthetic feasibility, considering various non-standard structural features such as large rings, unconventional ring fusions, stereocomplexity, and overall molecule size (Ertl & Schuffenhauer, 2009). However, diffusion-based generative models face a limitation in accurately positioning atoms, leading to unrealistic molecular topologies, especially the formation of large rings, which can negatively impact SA scores (Peng et al., 2023).

As evident from Table 10, the diffusion-based models exhibit a subpar distribution of ring sizes, notably characterized by a prevalence of larger rings. In TAGMOL, the provision of multiple guidance signals to coordinates increases the likelihood

of forming large rings. Consequently, this propensity towards larger ring formations contributes to either a drop in SA values or minimal improvement despite the guidance provided.

*Table 10.* Distribution of ring sizes in reference and generated molecules, expressed as percentages

| Ring Size | Ref | Pocket2 Mol | Target Diff | Decomp Diff | TAGMOL |
|---|---|---|---|---|---|
| 3 | 1.7% | 0.1% | 0.0% | 2.9% | 0.0% |
| 4 | 0.0% | 0.0% | 2.8% | 3.7% | 2.4% |
| 5 | 30.2% | 16.4% | 30.8% | 30.8% | 26.9% |
| 6 | 67.4% | 80.4% | 50.7% | 45.6% | 48.6% |
| 7 | 0.7% | 2.6% | 12.1% | 11.6% | 15.0% |
| 8 | 0.0% | 0.3% | 2.7% | 2.3% | 3.7% |
| $\geq 9$ | 0.0% | 0.2% | 0.9% | 3.1% | 3.4% |

## G. Pseudo Code for TAGMOL

This section provides a summary of the overall sampling procedures employing multiple guidances.

---
**Algorithm 1** Psuedo Code of TAGMOL
---
**Require:** The protein binding site $\mathbf{P}$, generative model $\theta$, Property Predictors $\phi_{\mathbb{Y}}$.
**Ensure:** Generate ligand molecule $\mathbf{M}$ that binds to the protein pocket & optimized for properties $\mathbb{Y}$
    Sample $N_{\mathcal{M}}$ atoms based on a prior distribution relative to the pocket size.
    Move CoM of protein atoms to zero.
    Sample coordinates $\mathbf{X}_T^{\mathcal{M}}$ and atom types $\mathbf{V}_T^{\mathcal{M}}$ based on prior:
    $\mathbf{X}_T^{\mathcal{M}} \sim \mathcal{N}(0, \mathbf{I})$
    $\mathbf{V}_T^{\mathcal{M}} = \texttt{one\_hot}(\arg\max_i g_i)$, where $g \sim \text{Gumbel}(0, 1)$
    **for** $t = T, T - 1, \ldots, 1$ **do**
        Calculate $[\widehat{\mathbf{X}}_{0|t}^{\mathcal{M}}, \widehat{\mathbf{V}}_{0|t}^{\mathcal{M}}]$ using $\theta([\mathbf{X}_t^{\mathcal{M}}, \mathbf{V}_t^{\mathcal{M}}], t, \mathcal{P})$
        Calculate $\tilde{\mu}_t\left(\mathbf{X}_t^{\mathcal{M}}, \hat{\mathbf{X}}_{0|t}^{\mathcal{M}}\right)$ according to posterior in Equation 13
        Calculate $\delta$ according to Equation 10
        Sample $\mathbf{M}_{t-1}$ according to the Equation 9
    **end for**
---

## H. Time Complexity

When executed on an NVIDIA A100 GPU, we observed a notable processing time trend for inference. Guidance for properties like QED and SA led to significant improvements (refer to Table 6) with minimal time increases of 1.06x and 1.07x, respectively, compared to the backbone. However, guiding for Binding Affinity (BA) led to a more substantial time increase due to the inclusion of protein atoms, demanding extra computational effort compared to QED or SA guidance, which involves only ligand atoms. Ultimately, TAGMOL recorded its longest processing time of 1.87x when applying guidance for all properties. It is important to note that TAGMOL is still comparable to 1.72x of DecompDiff (Guan et al., 2023a) sampling time.

*Table 11.* Inference time for different models

| Method | Time (s) |
|---|---|
| backbone | 938 |
| backbone + BA opt | 1646 |
| backbone + QED opt | 995 |
| backbone + SA opt | 1007 |
| TAGMOL | 1755 |

## I. Limitations and Future Work.

Although the present work shows promising results, we share some complementary directions worth investigating in the future. First, improved representation of the system using additional inductive bias in terms of symmetry and coarse-graining could furnish more accurate predictions of different properties that are structure dependent. Second, model compression and parameter pruning could speed-up inference without compromising performance. Third, integrating constraints on discrete attributes into gradient-based optimization may enable the modeling of complementary properties, such as enforcing hydrogen bonds between specific residues within the pocket, a capability our current algorithm lacks. Fourth, exploring reduced noise sampling strategies or guidance mechanisms for generating more precise conformations could lead to reduced strain energy and the production of stable molecules. Finally, it would be worth investigating physics-based techniques, such as molecular simulations, to allow generalization to larger family of systems.