# OpenReview forum: "TAGMol: Target-Aware Gradient-guided Molecule Generation"
_ICML.cc/2024/Workshop/ML4LMS — ML4LMS Poster_

### Official Review · Reviewer_DB88 · 2024-06-10
**Bold new approach to the drug discovery problem with a very interesting property prediction-guided conditional diffusion molecular generation model**

**Rating:** 9
**Confidence:** 3

**Review:**

**Summary:** this paper introduces a bold new approach to the drug discovery problem, by using the gradients of a property prediction model to guide the generation of new molecules, as a way to improve performance of generative approaches in complex multi-objective setups.

**Strengths:**
- Well written paper outlining a bold new vision for drug discovery supported by a well presented algorithmic and training setup, and well documented experimental evidence.
- The paper introduces clear and approachable theoretical setup despite the complexity of the approach, and motivates architecture choices well.
- Guide training is an original approach which makes great use of existing building blocks to introduce a novel and exciting approach.
- The setup leads to significant results, which could prove to be a seminal contribution to the field if results generalize.

 **Weaknesses:**
-  Previous works on latent diffusion models for molecular generation are missing, including recent state of the art results such as Xu et al.'s GeoLDM equivariant geometric latent diffusion, itself building on previous work in the field.

**Questions & Limitations:**
- **Discussing experimental performance:** the central claim of the paper is to improve drug discovery by guiding the generation process thanks to the gradients of a property prediction model. While the obtained Vina scores are indeed impressively better than other models, these baselines outperform TAGMol on property prediction (QED, SA), and diversity scores. It would be helpful to explain why that is the case? Why would improved binding affinity scores adequately compensates for less optimal performance on molecular properties? Why is that the case given that property prediction is used to guide generation?

**Minor typos/edits:**
- Line 149/150 right column: consider using \citet{} in latex to avoid extraneous parentheses.
- Line 245-250-252: minor consistency in typography for the s^{opt} variable.
- Consider moving more equations (e.g. 5 or 6) to appendix to improve general readability.
- Table 1 is a great summary, which may be made significantly more readable by using color highlights instead of bold and underling (but this is very subjective).

---

### Official Review · Reviewer_1oYN · 2024-06-12
**Application of Classifier Guidance to Structure-Based Drug Design**

**Rating:** 6
**Confidence:** 4

**Review:**

This paper proposes TAGMol, a method for target- and property-conditioned 3D molecule generation, and applies it to structure-based drug design. The approach uses standard classifier guidance: an SE(3)-invariant graph neural network is trained to predict properties like binding affinity, drug-likeness, and synthetic accessibility, and its gradients are used to steer the sampling process of a pre-trained unconditional diffusion model towards molecules with desired properties. The authors demonstrate that this outperforms a range of unconditional baselines on the CrossDocked2020 dataset.

The paper is well-written and clearly outlines the authors' approach. Since it is mainly an application of standard conditioning techniques to pre-existing diffusion models for SBDD, it might be appropriate to tone down some of the novelty claims. In the future, comparing the model to other property-aware molecular generation techniques, rather than just unconditional models, would help to further establish the benefits of the presented approach.